# Mechanism of gene network in the treatment of intracerebral hemorrhage by natural plant drugs in Lutong granules

**Jie Sun[1,2], Na Li[3], Min Xu[4]‡, Li Li[5]‡, Ji Lin Chen[3], Yong Chen[3], Jian Guo Xu[1]\*, Ting Hua Wang[1,3,4]\***

**1** Department of Neurosurgery, West China Hospital of Sichuan University, Chengdu, Sichuan Province, China, **2** Department of Neurosurgery, The first Hospital of Kunming, Kunming, Yunnan Province, China, **3** The Institute of Neuroscience, Kunming Medical University, Kunming, Yunnan Province, China, **4** School of Basic Medicine, Jinzhou Medical University, Jinzhou, Liaoning Province, China, **5** Department of acupuncture, Kunming Municipal Hospital of Traditional Chinese Medicine, Kunming, Yunnan Province, China

 These authors contributed equally to this work.
‡ MX and LL also contributed equally to this work.
\* wangtinghua@vip.163.com (THW); jianguo_1229@126.com (JGX)

**Data Availability Statement:** All relevant data are within the article and its Supporting information files.

**Funding:** This study was supported by the Science and Technology Major Project of Sichuan Province

## Abstract

### Purpose

To study the effects of Lu-tong Granules (LTG) in ICH etermine the underlying mechanism of molecular network

### Methods

Modern bioinformatics and network pharmacology methods were used to predict molecular network mechanisms between ICH and LTG. Animal experiments were carried out to verify the effect of LTG for the treatment of ICH, combined with behavior test and morphologic detection.

### Results

Forty-three active components in LTG and involved 192 gene targets were identified successfully. Moreoner, they were intersected with 1132 genes of ICH,88 intersection targets were obtained. subsequently, Cytoscape was used to screen Hub genes, in which,6 core molecules, including AKT1, IL6, VEGFA, CASP3, JUN and MMP9 were recognized. Furthermore, we constructed Six core compounds by " disease-drug-active ingredient-target-KEGG " (D-D-A-T-K) network, showed including quercetin, luteolin, β sitosterol, stigmasterol, kaempferol and formononetin, and PPI protein network interaction showed that AKT1: OS3 and CNA2:DKN1A had the highest correlation. Whereas the enrichment of GO and KEGG indicated that LTG was most likely to play a therapeutic role in ICH through AGE-RAGE signaling pathway in diabetic complications. Integrated analysis also showed that the first 10 pathways of KEGG are integrated into 59 genes, among which 6 core genes are closely involved. Lastly, molecular docking showed that there was a good binding activity

(Project No. 2020YFS0043), Kunming Science and Technology Plan (Project No 2020-1-H-024).The funders had no role in study design, data collection and analysis, decision to publish, or preparation of the manuscript.

**Competing interests:** We confirm that the manuscript has been read and approved by all the undersigned authors. We confirm that we have given due consideration to the protection of intellectual property associated with this work and that there are no impediments to publication, including the timing of publication, with respect to intellectual property. In so doing we confirm that we have followed the regulations of our institutions concerning intellectual property. We understand that the Corresponding Author is the sole contact for the Editorial process (including Editorial Manager and direct communications with the office). He/she is responsible for communicating with the other authors about progress, submissions of revisions and final approval of proofs. We confirm that we have provided a current, correct email address which is accessible by the Corresponding Author and which has been configured to accept email from PLOS ONE. This does not alter our adherence to PLOS ONE policies on sharing data and materials.

between the core components and the core genes, and animal experiments confirmed effect of LTG in the treatment of ICH, by using TTC staining and behavior test.

## Conclusion

LTG are effective for the treatment of ICH, the underlying mechanism could be involved in gene network including anti-inflammatory response, nerve repair, analgesia, anti-epilepsy and other aspects.

## 1. Introduction

ICH accounts for about 10–20% of all cerebral apoplexy, with high mortality and disability rates [1]. The core of current treatment strategies for ICH is to improve survival and neurological outcomes, the selection of the optimal medication and timing of operation for ICH [2] is there for important and it keeps still in the exploratory stage. For the primary mechanical injury of ICH, the objective of surgery is to remove the hematoma, reduce the intracranial pressure, and reduce the secondary damage [3]. Studies have shown that early surgery may be beneficial [2, 4]. Subsequently, secondary damage to ICH is mediated by a variety of complex mechanisms, mainly including toxic products and inflammatory reactions [5]. The aim of control edema around ICH hematoma is conducive to alleviate neurological dysfunction and promoting rehabilitation [6]. As the complexity and variability of ICH, as well as the numerous and the effect is limit affecting factors, individual diagnosis and treatment choices becomes more difficult. To find specific effective drug and explore gene mechanism effective intervention of ICH keeps on the challegible way.

TCM has evolved from historical practice and form systematic theoretical basis. The formula of TCM has the characteristics of multi-system, multi-component and multi-target in the diagnosis and treatment of diseases. But its medicinal mechanism is not clear yet. LTG is a Specific product, based on modern Chinese medicine innovation theory and developed according to clinical practice experience. This prescription has been clinically used for the treatment of migraine and other neurological diseases. Its ingredients include Notopterygium incisum, Kudzu root, Schizonepeta, Ligusticum chuanxiong, Ligusticum sinense Oliv, Fructus viticis, Scorpion. Although the natural botanicals and active ingredients in the formulations have been reported in the literature, the network mechanism in ICH is largely unknown. Therefore, experiment combined with modern bioinformatics, network pharmacology and other methods, we systematically evaluate the effect of LTG in ICH and explore its active ingredients, potential targets, signaling pathways so as to provide data support for LTG application in ICH treatment. We have provided a mind map (Fig 1).

## 2. Materials and methods

### 2.1 Collection and format conversion of active ingredients and targets of LTG

We retrieved LTG from the Traditional Chinese Medicines Systems Pharmacology Platform (TCMSP). The active ingredients and target of each plant drug were screened based on the oral bioavailability (OB) $\geq$ 30 and drug-likeness (DL) $\geq$ 0.18. And then the UniProt database (https://www.uniprot.org/) is used to check and convert Gene names.

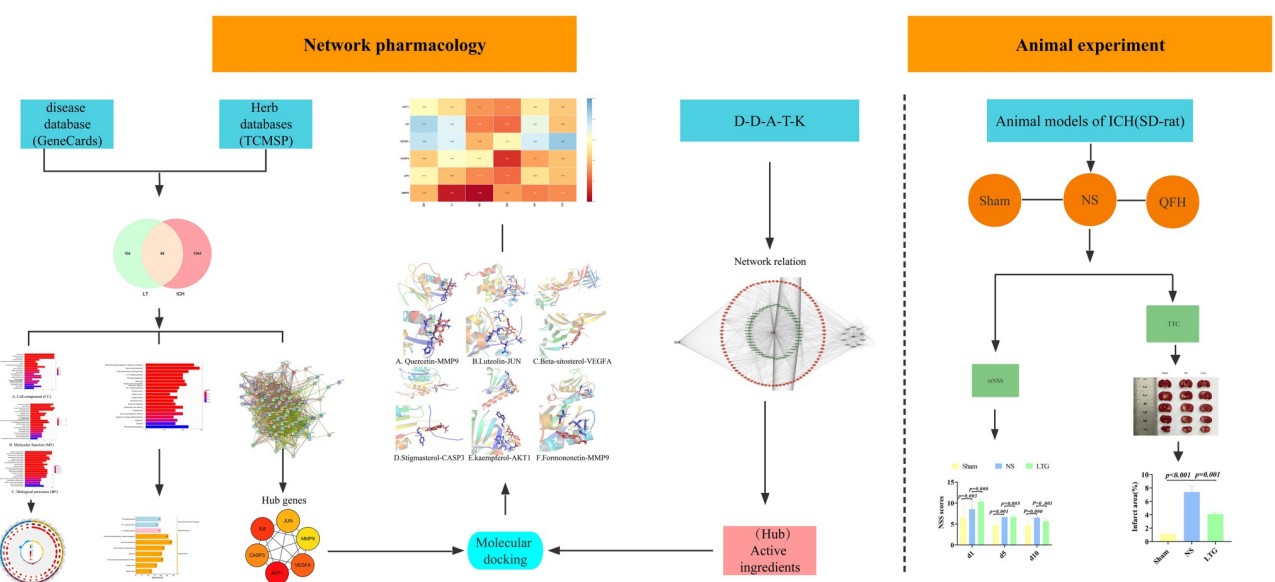

**Fig 1. Mind map.** Note: Modern bioinformatics combined with network pharmacology techniques were applied to explore the mechanism of action of LTG on ICH in conjunction with animal experiments.

## 2.2 Identification of ICH related targets

We searched the keyword "ICH" to get all genetic data and download them in the GeneCards database (https://www.genecards.org).

## 2.3 The intersection of drug active ingredient target and ICH target

Venny 2.1 database (https://bioinfogp.cnb.csic.es/tools/venny/) has been applied to get the intersection of LTG active ingredient target and ICH related target, and draw the Venn diagram.

## 2.4 Gene Ontology (GO) and KEGG signaling pathway classification

We mapped the intersection data to the Geneontology database (http://www.geneontology), and GO (Cellular Component, CC), (Molecular function, MF) and (biological process, BP) were analyzed by using the whole human genome as background gene and reference set, and histogram and pathway circle diagram were output, lastly. Then KEGG signal transduction pathway was extracted (http://www.genome.jp/kegg/), and the histogram and classification diagram were drawn.

## 2.5 Construction of Protein Protein Interaction (PPI) network map

Venn intersection data of drug active ingredients and ICH key targets were uploaded to String database (https://string-db.org/), and species were defined as "Homo Sapiens", and PPI network interaction diagrams were drawn. Next, Cytoscape3.7.2(http://www.Cytoscape.org/) software is used to screen Hub genes and perform visualization and network topology analysis on the resulting data.

## 2.6 Construction of " D-D-A-T-K " network diagram

After the " D-D-A-T-K " data are input into Cytoscape3.7.2 software, analyzer Network tool is used to calculate and select the core active ingredients according to degree value to build a Network graph.

## 2.7 Docking verification of main active ingredients and key gene target molecules

The 6 core targets were screened out for molecular docking with 6 core active components. In detail Six drug effective components were input in PubChem database (https://pubchem.ncbi.nlm.nih.gov), which in turn to get sdf molecular structure of 2D format. Openbabel-2.4.1 converts SDF to MOL2 format for backup. In the PDB database (https://www.rcsb.org), "Homo sapiens" and REFINEMENT RESOLUTION which less than 2.0 were used as screening conditions and protein structures were downloaded. Water molecules and ligands in the macromolecule protein structure were removed with PyMol software, and the protein structure was hydrogenated. Gasteiger charge was calculated and the AD4 type was set after energy minimization. AutoDock VINa1.1.2 was used for molecular virtual docking, and the binding conformation with low free binding energy was selected for virtual docking screening and heat map drawing. The model with the lowest binding energy was visualized by using PyMol2.3.4 software.

## 2.8 Animal experiments

**2.8.1 Experimental animals, grouping and drug preparation.** Adult male Sprague-Dawley (SD) rats with body weight between 200 and 220 g were selected and provided by the Department of Experimental Animal Science, Kunming Medical University (Ethics No KMMU20220852, batch No. SCXK Yunnan K, 2020–0004). They were adaptively fed for 1 week at room temperature of 20 to 25°C according to the circadian rhythm.

Animals and grouping

SD rats were randomly divided into three groups: ① Sham group (n = 10); ② NS group (n = 10), ③ LTG group (n = 10);

LTG is approved by Sichuan Food and Drug Safety Monitoring and Certification Center, prepared by the Preparation room of The Affiliated Hospital of Traditional Chinese Medicine of Southwest Medical University. The ingredients condidts of notopterygium incisum 200 g, kudzu root 300 g, Schizonepeta 150 g, ligusticum chuanxiong 300 g, Ligusticum sinense Oliv 220 g, fructus viticis 150 g, scorpion 70 g, melted Soak in 7 times the amount of water for 0.5 hours and cook twice for 1 hour each, incorporating the filtrate, in water with 100°C. to concentrate to a clear paste with a relative density of 1.20–1.25 (80°C). lastly 700 g of dextrin was added to make into granules (10 g/ bag). The extract yield of Lutong granules is 25%, labelled with batch number: 20190829, 4.0 g/kg/d in dose, administrated at gavage.

**2.8.2 The experimental process.** Before the experiment, SD rats were deprived of food and water for 24 hours and anesthetized with 3% sodium pentobarbital, based on body weight. To perform operation SD rats were fixed in prone position on stereotactic apparatus. After routine disinfection of the scalp, the skull was positioned 2.5 mm to the right of the midline of the anterior fontanelle. The needle was moved 2.0 mm backward along the sagittal plane, and the insertion depth was 4.5 mm. After slow injection of caudal Venous blood about 50 µL (The sham group does not inject blood), the needle was held for 10 min to prevent blood overflow. The bone wax closed the bone foramen after the needle was slowly removed. The scalp was sutured and disinfected. During the experiment, the usage and pain of animals should be minimized. After the behavioral experiment, SD rats were sacrificed by cervical dislocation.

**2.8.3 Modified Neurological Severity Scores (mNSS) and TTC staining procedure.** On days 1 to 10 postoperatively, a modified mNSS score was performed to assess neurological deficits in motor, sensory, balance, and reflexes. In order to ensure the accuracy of experimental data as far as possible, 3 experimenters were required to score without communication and take the mean value. The data were collected and statistically analyzed.

The TTC (2,3,5-triphenyltetrazolium chloride) staining group was performed on day 7. In detail, animals were anaesthetised with 3% sodium pentobarbital, the brains were quickly removed and transferred in PBS solution at 0–4˚C, then frozen in a refrigerator at -20˚C for 30 min. Subsequently, coronal brain slices were cut to a thickness of 2 mm, then placed in a 2% solution of TTC and soaked for 30 min at 37˚C, protected from light, and keep in 4% paraformaldehyde for 12 h. Lastly, digital photography was performed, and infarct volume analysis was calculated by an image analysis program (ImageJ® software). Infarct volume (%) = (area of contralateral hemisphere—healthy area of ipsilateral hemisphere) × thickness of brain slice.

# 3. Results

## 3.1 Collection and screening of active ingredient targets of LTG

The active ingredients for LTG were collected by TCMSP and 11 kinds of notopterygium incisum, 10 kinds of Schizonepeta, 6 kinds of Ligusticum chuanxiong, 1 kind of ligusticum sinense, 23 kinds of fructus viticis and 3 kinds of kudzu root were collected. After integration, we aquired a total of 43 kinds of active ingredients were obtained. Meanwhile, a total of 457 target genes matched with active ingredient, including 34 notopterygium incisum, 168 Schizonepeta, 22 Ligusticum chuanxiong, 3 ligusticum sinense, 183 fructus viticis and 47 kudzu roots were collected. After removing the duplicate genes, they were integrated into 192 targets.

## 3.2 Collection of ICH related targets

1132 related targets were found in the database of GeneCards (https://www.Genecards.org) with the keyword "ICH" and they were downloaded into the local Excel table.

## 3.3 Intersection of drug active ingredients with key ICH targets

After the intersection of LTG and ICH related targets, 88 key targets were obtained, and Venn diagram was drawn, and exhibited in (Fig 2A).

## 3.4 Gene ontology (GO) and KEGG signaling pathway analysis

To further investigate the mechanism of action of LTG in the treatment of ICH, we performed GO and KEGG (Fig 3A) enrichment analysis of the intersecting genes. The results showed that a variety of cellular components, biological processes and molecular functions were involved (Fig 2C–2E). In which, membrane raft, protein kinase complex, vesicle lumen and other cellular components, as well as processes such as response to lipopolysaccharide and response to oxidative stress, were involved. There are 59 genes integrated in the first ten KEGG pathways, of which, six Hub genes are distributed in different pathways (Table 1). The first pathway, AGE-RAGE signaling pathway in diabetic complications (hsa04933) (Fig 3C), was enriched for five Hub genes, namely JUN, CASP3, AKT1, VEGFA and L6. Oxidative stress can cause AEGs to activate the receptor RAGE on the cell membrane, which acts indirectly on AKT, AP-1 and NF-κB through PI3k-Akt signaling pathway and calcium signaling pathway, activating IL-1, IL-6, IL-8, VEGFA, CASP3, thereby inducing a series of processes such as inflammation, apoptosis and endothelial production. In the graph of the first ten GO-Pathways enrichment circles (Fig 2B), the first circles are BP (yellow), CC (purple) and MF (blue) with codes. We

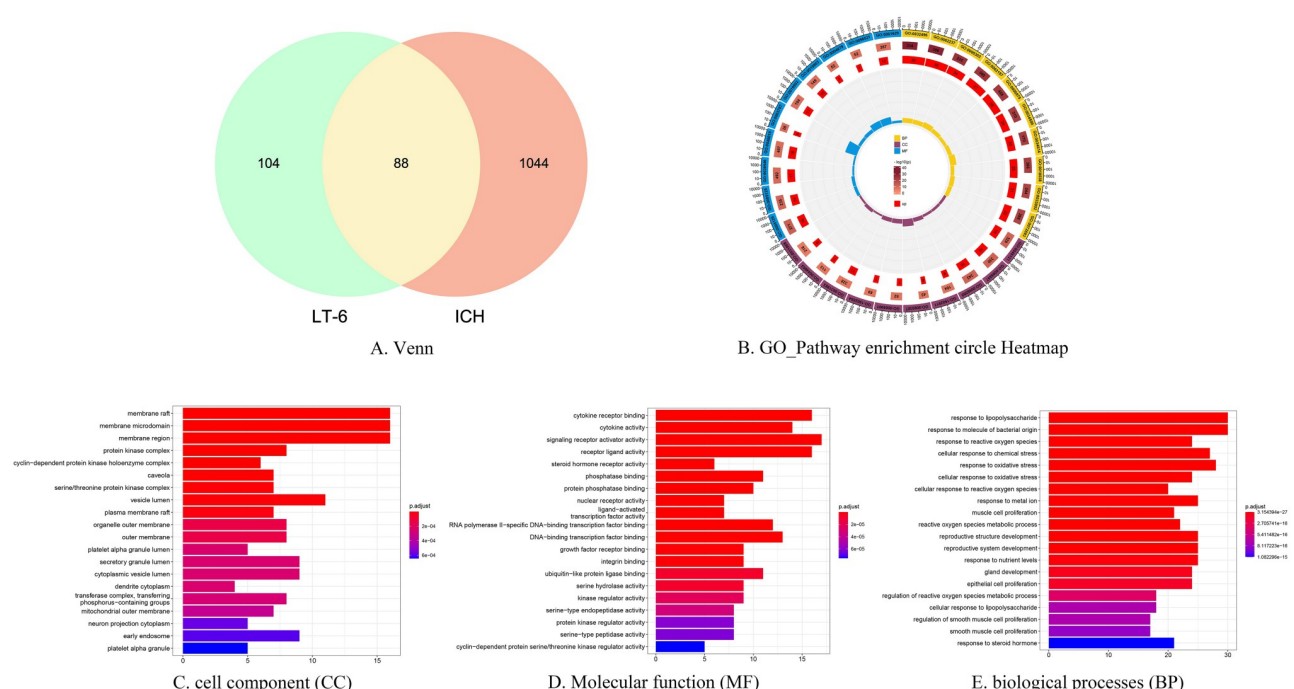

**Fig 2. Venn diagram and GO enrichment analysis.** Note: 2A Venn diagram of active ingredient targets and ICH targets. 2B GO-Pathway enrichment circle Heatmap. 2C Cell components (CC) 2D Molecular function (MF) 2E Biological processes (BP).

ranked the genes according to the number of enriched genes, selected the top three and presented them in the table (Table 2). The second circle is the total number of terms enriched in the pathway. The third circle is the number of upregulated genes in the term. The inner circle is the enrichment factor, and the results show multiple regulated genes enriched in the top ten pathways of BP, CC and MF. It is hypothesized that LTG acts on ICH through the relevant pathways and genes mentioned above. The classification of the top ten KEGG pathways showed that they were concentrated in three main categories, namely environmental information processing, organismal systems, and human diseases (Fig 3B).

## Example of GO-pathway enrichment circular diagram

Table 1.

## List of genes in the top 10 KEGG pathways

Table 2.

## 3.5 Construction of protein-protein interaction network diagram of intersection targets

After the 88 "intersection targets" of LTG and ICH were identified as "Homo Sapiens" in String database (https://string-db.org/), PPI network interaction was mapped (Fig 4A). We found that there was a correlation between these 88 targets. According to combined_SCORE numerical ranking, the top ten targets of connection strength (Table 3) are: AKT1:NOS3; CCNA2: CDKN1A; CCND1:CDKN1A; CCND1:ESR1; CDKN1A:PCNA; EGF:ERBB2; EGF:EGFR; ESR1:JUN; F3:F7; HIF1A:MDM2; Through the analysis of the obtained data files, we find 88 nodes, and 1468 edges as well as, the average node degree is 33.4; Average local clustering

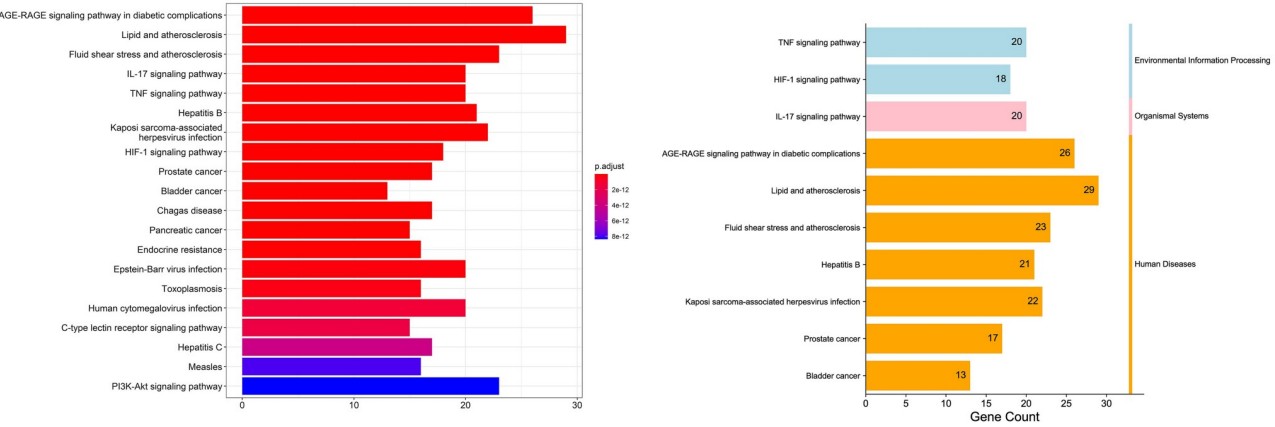

A. KEGG signal pathway diagram

B. Classification diagram of enrichment result

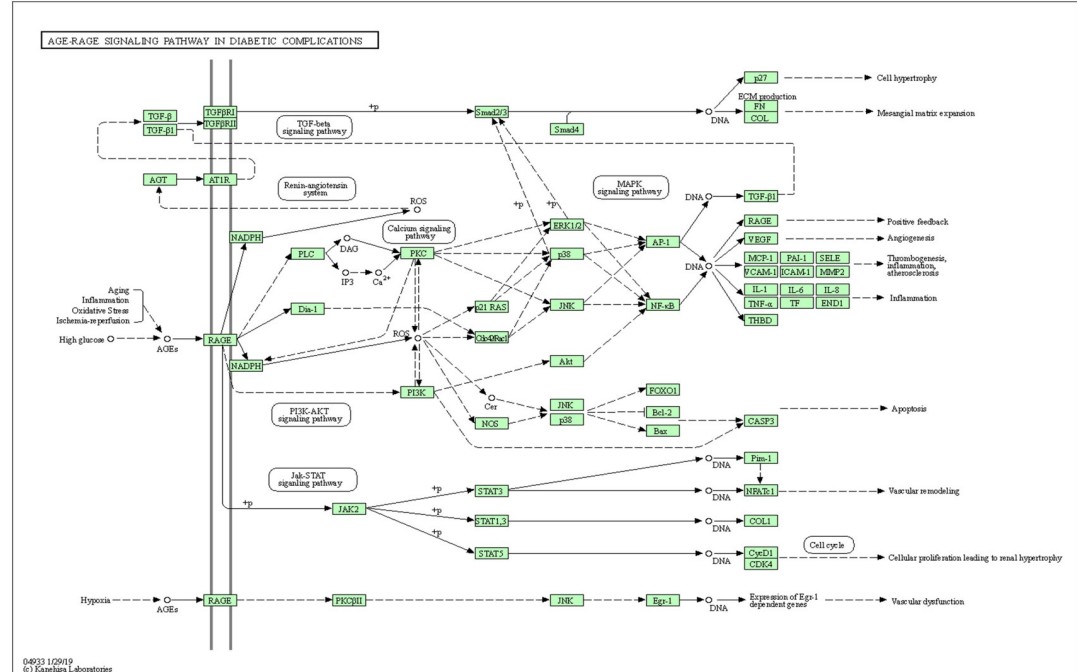

C. AGE-RAGE signaling pathway in diabetic complications pathway

**Fig 3. Pathway diagram in KEGG pathway analysis.** Note: The ordinate of 3A is the first 20 KEGG pathways enriched, and the colors of 3B represent different classifications. 3C AGE-RAGE signaling pathway in diabetic complications pathway (KEGG ID = Hsa04933).

coefficient is 0.73; expected edge number is 388; PPI enrichment P value is less than 1.0 e-16. lastly Through Cytoscape3.7.2(http://www.Cytoscape.org/) software, six Hub bases are screened out according to drgee values, which are AKT1, IL6, VEGFA, CASP3, JUN and MMP9 (Fig 4B). As 6 genes were distributed in the top ten LEGG pathways (Table 1), it indicates that "LTG" may act on ICH mainly through these core targets again.

## 3.6 Construction of (D-D-A-T-K) network and core component screening

The data of "drug, active ingredient, intersection target and KEGG pathway" were input into Cytoscape3.7.2 software and measured by Analyzer Network. According to degree value,

**Table 1. Table of the top 10 KEGG signaling pathways.**

| ID | Description | p.adjust | Matched target |
|---|---|---|---|
| hsa04933 | AGE-RAGE signaling pathway in diabetic complications | 4.45E-28 | BCL2, BAX, *JUN*, *CASP3*, RELA, *AKT1*, *VEGFA*, CCND1, MMP2, MAPK1, *L6*, ICAM1, STAT1, F3, IL1B, CCL2, SELE, VCAM1, CXCL8, NOS3, THBD, SERPINE1, IL1A, COL3A1, MAPK14, MAPK8 |
| hsa05417 | Lipid and atherosclerosis | 3.81E-23 | RXRA, BCL2, BAX, CASP9, UN, CASP3 CASP8, RELA, *AKT1*, *MMP9*, MAPK1 , *IL6*, MMP1, PPARG, ICAM1, CD40LG, MMP3, IL1B, CCL2, SELE, VCAM1, CXCL8, NOS3, NFE2L2, CXCL2 CHUK, MAPK14, MAPK8, APOB |
| hsa05418 | Fluid shear stress and atherosclerosis | 3.91E-20 | BCL2, *JUN*, RELA, *AKT1*, *VEGFA*, MMP2, *MMP9*, HMOX1, ICAM1, IFNG IL1B, CCL2, SELE, VCAM1, NOS3, PLAT, THBD, IL1A, NFE2L2, CHUK, KDR, MAPK14, MAPK8 |
| hsa04657 | IL-17 signaling pathway | 9.84E-20 | PTGS2, *JUN*, CASP3, CASP8, RELA, *MMP9*, MAPK1, *IL6*, MMP1, IFNG, IL4, MMP3, IL1B, CCL2, CXCL8, CXCL2 CXCL10, CHUK, MAPK14, MAPK8 |
| hsa04668 | TNF signaling pathway | 3.31E-18 | PTGS2, *JUN*, *CASP3*, CASP8, RELA, *AKT1*, *MMP9*, MAPK1, *IL6*, ICAM1, MMP3, IL1B, CCL2, SELE, VCAM1, CXCL2, CXCL10, CHUK, MAPK14, MAPK8 |
| hsa05161 | Hepatitis B | 2.98E-16 | BCL2, BAX, CASP9, *JUN*, *CASP3*, CASP8, RELA, *AKT1*, CDKN1A, *MMP9*, MAPK1, RB1, *IL6*, PCNA, BIRC5, STAT1, CXCL8, CHUK, MAPK14, CCNA2, MAPK8 |
| hsa05167 | Kaposi sarcoma-associated herpesvirus infection | 7.05E-16 | PTGS2, BAX, CASP9, *JUN*, *CASP3*, CASP8, RELA, *AKT1*, *VEGFA*, CCND1, CDKN1A, MAPK1, RB1, *IL6*, ICAM1, HIF1A, STAT1, CXCL8, CXCL2, CHUK, MAPK14, MAPK8 |
| hsa04066 | HIF-1 signaling pathway | 7.80E-16 | BCL2, NOS2, RELA, EGFR, *AKT1*, *VEGFA*, CDKN1A, *MAPK1*, *IL6*, ERBB2, HMOX1, IFNG, INSR, EGF, HIF1A, NOS3, SERPINE1, HK2 |
| hsa05215 | Prostate cancer | 2.03E-15 | BCL2, CASP9, RELA, EGFR, *AKT1*, CCND1, CDKN1A, *MMP9*, MAPK1, RB1, MDM2, ERBB2, MMP3, PLAU, EGF, PLAT, CHUK |
| hsa05219 | Bladder cancer | 2.25E-15 | EGFR, *VEGFA*, CCND1, CDKN1A, MMP2 , *MMP9*, MAPK1, RB1, MDM2, MMP1 ERBB2, EGF, CXCL8 |

Note: List of genes in the first 10 pathways of KEGG, where the Genes marked in bold italics represent the distribution of 6 HUB genes in each pathway.

quercetin, luteolin, β sitosterol, stigmasterol, kaempferol and formononetin were screened as the main active ingredients. And then find the source of each component (Table 4). After the "ICH" factor was added, the (D-D-A-T-K) network diagram was constructed (Fig 4C).

## 3.7 Docking of main active ingredients and key target molecules

Six core targets, namely AKT1, IL6, VEGFA, CASP3, JUN and MMP9, were recognized for molecular docking with six core compounds, namely quercetin, luteolin, β sitosterol, stigmasterol, kaempferol and formononetin. According to the 6 models with small binding values, we used PyMOL2.3.2 to show their binding states (Fig 5A and 5B).

**Table 2. Example of go-pathway enrichment circular diagram.**

| goterm | Description | category | totalnumber | Termnumber | pvalue | up_regulated | rich_factor |
|---|---|---|---|---|---|---|---|
| GO:0098589 | membrane region | CC | 19559 | 343 | 2.61E-12 | 16 | 0.046647 |
| GO:0098857 | membrane microdomain | CC | 19559 | 330 | 1.45E-12 | 16 | 0.048485 |
| GO:0045121 | membrane raft | CC | 19559 | 329 | 1.38E-12 | 16 | 0.048632 |
| GO:0030546 | signaling receptor activator activity | MF | 18352 | 492 | 1.54E-10 | 17 | 0.034553 |
| GO:0030546 | signaling receptor activator activity | MF | 18352 | 492 | 1.54E-10 | 17 | 0.034553 |
| GO:0048018 | receptor ligand activity | MF | 18352 | 487 | 1.18E-09 | 16 | 0.032854 |
| GO:0032496 | response to lipopolysaccharide | BP | 18866 | 334 | 8.82E-31 | 30 | 0.08982 |
| GO:0006979 | response to oxidative stress | BP | 18866 | 458 | 5.14E-24 | 28 | 0.061135 |
| GO:0010038 | response to metal ion | BP | 18866 | 366 | 1.37E-22 | 25 | 0.068306 |

Note: Description is the corresponding explanation for each goterm

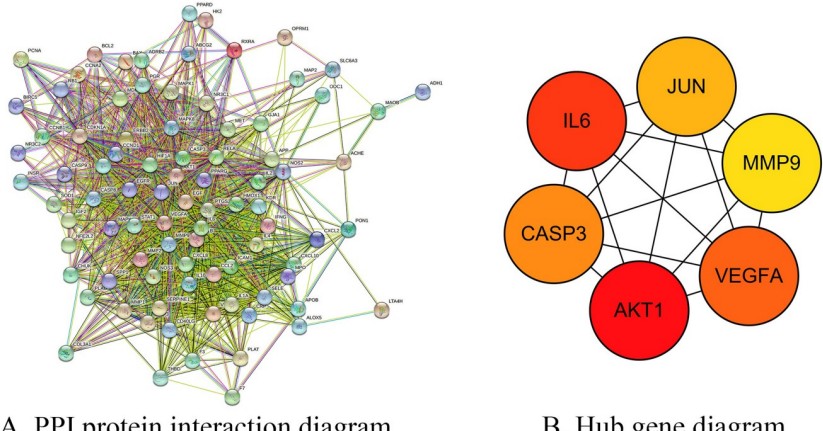

A. PPI protein interaction diagram

B. Hub gene diagram

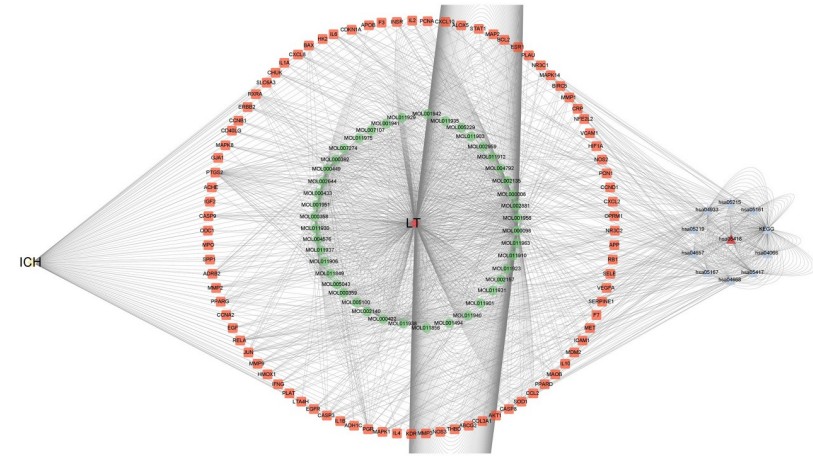

C. (D-D-A-T-K) network diagram

**Fig 4. PPI protein interaction map and Hub gene diagram and D-D-A-T-K.** Note: A. PPI protein interaction map of 88 intersecting genes. B. Hub gene diagram. C. (D-D-A-T-K) network diagram. ICH for disease, LTG for herb, HSA for KEGG pathway, orange for target, and green for drug active ingredient.

### 3.8 Findings of animal experiment

Modified mNSS score was performed on the 1st, 5th and 10th day after operation. The results of showed that there was an extremely significant difference between the NS group and the LTG group on the 1st postoperative day (P = 0.000). On postoperative day 5 (P = 0.005), and on postoperative day 10 (P = 0.001), there were a significant difference too. The results confirmed that LTG treatment of ICH in SD rats was superior to NS group in neurological function recovery. The results of repeated measurement anova show that the result of sphericity test is P = 0.114, P > 0.05, which satisfies the sphericity hypothesis. Based on the hypothesis sphericity, the results of intra-subject effect test showed time: P = 0.001, and inter-subject effect group: P = 0.052. (Fig 6, Table 5).

TTC staining was performed on the seventh postoperative day, and one-way Anova showed that the infarct area in the NS group was the largest, while that in the LTG group was smaller, (P = 0.001) (Fig 7A and 7B).

**Table 3. The connection strength ranked among the top ten pairs of targets.**

| node1 | node2 | node1_string_id | node2_string_id | coexpression | experimentally_determined_interaction | database_annotated | automated_textmining | combined_score |
|---|---|---|---|---|---|---|---|---|
| AKT1 | NOS3 | 9606.ENSP00000451828 | 9606.ENSP00000297494 | 0.049 | 0.879 | 0.9 | 0.988 | 0.999 |
| CCNA2 | CDKN1A | 9606.ENSP00000274026 | 9606.ENSP00000384849 | 0.061 | 0.921 | 0.8 | 0.972 | 0.999 |
| CCND1 | CDKN1A | 9606.ENSP00000227507 | 9606.ENSP00000384849 | 0.085 | 0.983 | 0.9 | 0.99 | 0.999 |
| CCND1 | ESR1 | 9606.ENSP00000227507 | 9606.ENSP00000405330 | 0 | 0.867 | 0.9 | 0.987 | 0.999 |
| CDKN1A | PCNA | 9606.ENSP00000384849 | 9606.ENSP00000368458 | 0.063 | 0.998 | 0.9 | 0.265 | 0.999 |
| EGF | ERBB2 | 9606.ENSP00000265171 | 9606.ENSP00000269571 | 0.16 | 0.321 | 0.9 | 0.989 | 0.999 |
| EGF | EGFR | 9606.ENSP00000265171 | 9606.ENSP00000275493 | 0.16 | 0.982 | 0.9 | 0.991 | 0.999 |
| ESR1 | JUN | 9606.ENSP00000405330 | 9606.ENSP00000360266 | 0 | 0.684 | 0.9 | 0.988 | 0.999 |
| F3 | F7 | 9606.ENSP00000334145 | 9606.ENSP00000364731 | 0 | 0.978 | 0.9 | 0.99 | 0.999 |
| HIF1A | MDM2 | 9606.ENSP00000437955 | 9606.ENSP00000258149 | 0.062 | 0.873 | 0.9 | 0.984 | 0.999 |

**Table 4. Core active ingredient screening list for "LTG".**

| name | Component name | Degree | Closeness Centrality | NeighborhoodConnectivity | Radiality | Stress | TopologicalCoefficient | source |
|---|---|---|---|---|---|---|---|---|
| MOL 000098 | quercetin | 544 | 0.526770294 | 4.518248175 | 0.993443 | 1038048 | 0.041883907 | Herba schizonepetae, Viticis Fructus |
| MOL 000006 | luteolin | 200 | 0.406125166 | 7.115384615 | 0.989326 | 354510 | 0.073679333 | Herba schizonepetae, Viticis Fructus |
| MOL 000358 | beta-sitosterol | 156 | 0.376078915 | 9.413793103 | 0.98789 | 326278 | 0.110707804 | Herba schizonepetae, notopterygium, Kudzu root |
| MOL 000449 | Stigmasterol | 104 | 0.377008653 | 9.785714286 | 0.987938 | 179204 | 0.112637363 | Herba schizonepetae, Viticis Fructus |
| MOL 000422 | kaempferol | 102 | 0.39869281 | 7.176470588 | 0.988991 | 323628 | 0.080213904 | Viticis Fructus |
| MOL 000392 | formononetin | 56 | 0.347776511 | 7.25 | 0.986311 | 233536 | 0.138888889 | Kudzu root |

Note: Source is the core ingredient in what natural plant medicine

## 4. Discussion

ICH is one of the great challenges in neurological diseases, and only 10–20% of survivors can live independently [7]. The incidence rate is 84% in middle-income or low-income countries [8]. Therefore, the search for effective treatment strategies for ICH is a worldwide public health problem. In the past few years, researchers have invested a lot of energy from the basic to the clinical. Extensive studies have been conducted on modifiable factors of ICH, except the non-modifiable factors such as age and bleeding site. Such as blood pressure level, blood glucose control, body temperature, coagulation function, hemoglobin toxicity, brain edema, intracranial pressure, hematoma enlargement and surgical methods. The objective was to reduce mortality after ICH and improve neurological outcomes. ICH hematoma may cause primary mechanical nerve injury. The ensuing eruption of inflammation and cerebral edema around hematoma, which are the core processing elements of the acute phase of ICH, can lead to more severe and lasting secondary damage. Inhibition of cerebral edema improves the treatment of ICH [9]. Two large clinical trials (Interact2, ATACH2) showed a beneficial trend in ICH's acute hypotension [10, 11]. With the rapid development of bioinformatics, network pharmacology and computer science, it has become a promising direction to find intervention targets for ICH at the genetic level. Traditional Chinese medicine has been developed for thousands of years by combining clinical practice with the theory of holistic dialectical thinking. Chinese herbal medicine is a natural substance that maintains the natural nature and biological activity of various ingredients and is easily absorbed by the human body with little side effects. The natural compounds found in herbal medicines have been an important source of discovery for new drugs. The use of botanical drugs to treat and prevent neurological diseases has a long history [12]. Many plant drugs are widely distributed around the world and are easy to grow and obtain. Studies have shown that phytochemicals with natural biological activity can regulate neurotrophic factors and contribute to nerve regeneration [13]. Based on the advantages of natural plant drugs in multi-orientation, multi-angle, multi-link, multi-level and multi-target, the research group selected "LTG" of notopterygium incisum, Kudzu-root, Schizonepeta, Ligusticum chuanxiong, ligusticum sinense and fructus viticis, and studied the efficacy, action target and synergistic network mechanism of "LTG" on "ICH" by using modern molecular biology and other multidisciplinary collaboration ideas.

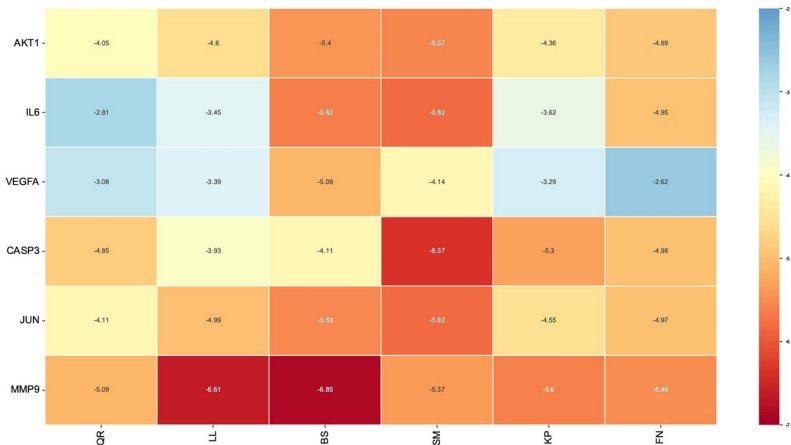

A. Molecular docking binding energy clustering heat map

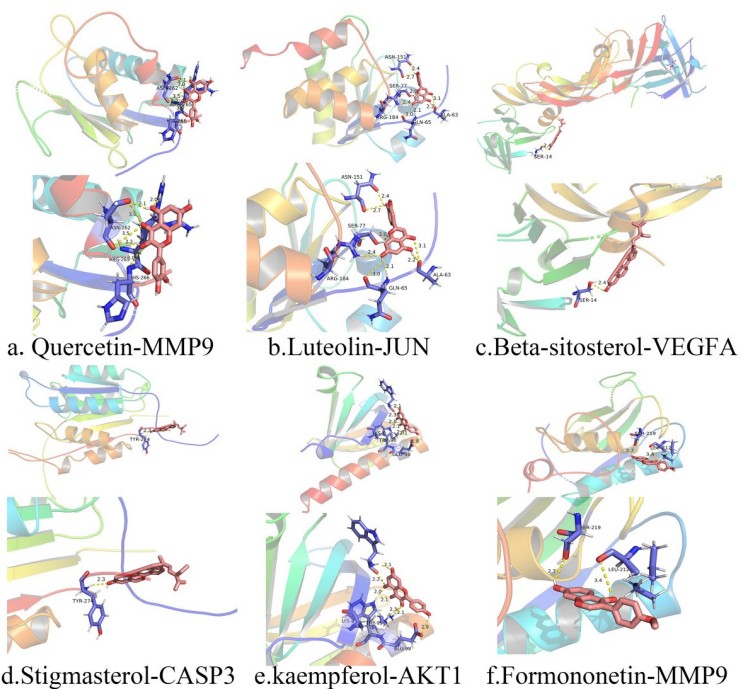

a. Quercetin-MMP9    b.Luteolin-JUN    c.Beta-sitosterol-VEGFA

d.Stigmasterol-CASP3    e.kaempferol-AKT1    f.Formononetin-MMP9

B. Docking effect drawing of core components and core targets

**Fig 5. Molecular docking diagram.** Note: A. Heat map of molecular docking binding energy. The smaller the value of molecular docking binding energy is, the docking is stable, and the color changes with the value of binding energy. Quercetin(QR), luteolin(LL), beta-sitosterol (BS), Stigmasterol (SM), kaempferol (KP), formononetin (FN). B. Docking effect drawing of core components and core targets.

In this study, Through TCMSP, 43 active ingredients for LTG corresponding to 192 target genes were identified, and quercetin belongs to flavonols, luteolin, kaempferol and formono-netin belongs to flavonoids, β sitosterol and stigmasterol belong to plant sterols. These natural compounds are found in a wide variety of plants and have powerful antioxidant and anti-

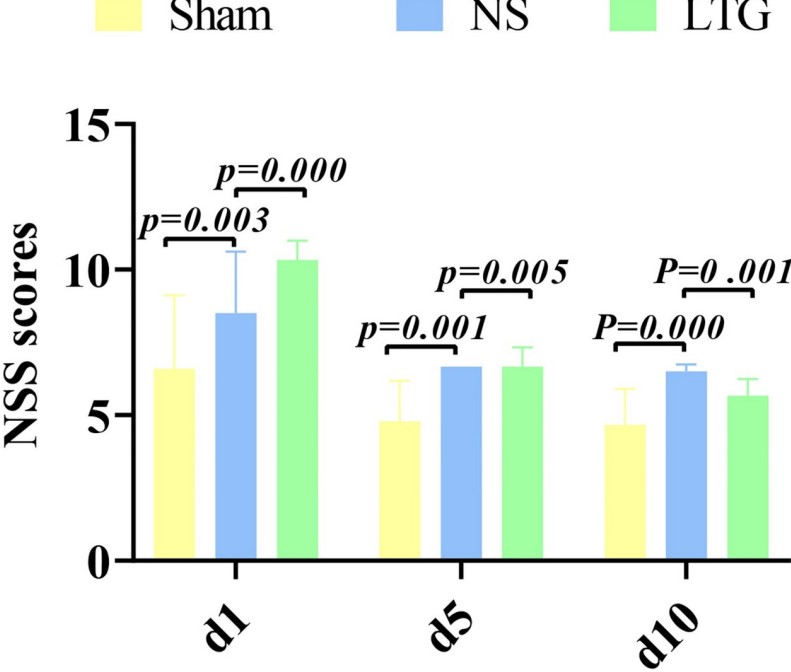

**Fig 6. Histogram of modified mNSS score.** Note: the abscissa is the postoperative days (d), and the ordinate is the improved Nss score.

inflammatory activities. After ICH, quercetin can repair nerve injury by inhibiting inflammatory response and cell apoptosis [14]. As one of the most active plant-based antioxidants, its efficacy in neurodegenerative diseases has been demonstrated [15, 16]. The pleiotropism of quercetin is also reflected in the lowering of blood pressure, antiviral, anti-cancer and cardiac protection, as well as the significant influence on the levels of high-density lipoprotein cholesterol and triglyceride [17] Studies have confirmed that 250 mg/day of type 2 diabetes for 8 weeks can improve the amount of insulin, blood glucose and lipid spectrum [18]. Luteolin and Kaempferol also have powerful anti-inflammatory and antioxidant effects and can inhibit proinflammatory mediators IL-6. In vivo and in vitro experiments of ICH model, luteolin can effectively relieve cerebral edema and inhibit neuronal degeneration in vivo to provide brain protection. In this process, Nrf2 ubiquitination and the production of neuronal mitochondrial superoxide (MitoSOX) are inhibited, and mitochondrial damage of neurons in vitro is reduced

**Table 5. Statistical scale of modified mNSS score.**

| Repeat measure ANOVA table | | | | | |
|---|---|---|---|---|---|
| **Variables** | **DF** | **SS** | **MF** | **F** | **P** |
| Intervene | 2 | 32.451 | 16.226 | 4.646 | 0.052 |
| Intergroup error | 7 | 24.449 | 3.493 | | |
| Time | 2 | 41.585 | 20.792 | 13 | 0.001 |
| Time*Intervene | 4 | 8.088 | 2.022 | 1 | 0.319 |
| Repeated measurement | 14 | 21.831 | 1.559 | | |

Note: DF: degree of freedom, SS: sum of squares, MF: mean square, P: significance

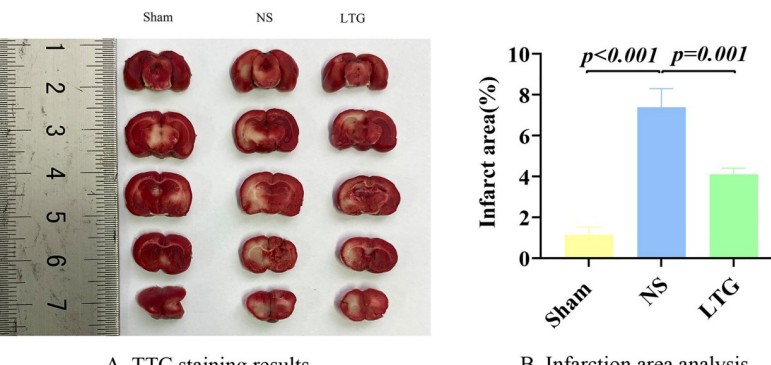

**Fig 7. TTC staining and infarct size calculation.** Note: A. TTC staining results. B. Statistical analysis of infarct size.

[19, 20]. It is noteworthy that flavonoids have similar structures to benzodiazepines and can be developed as an alternative to antiepileptic drugs. Kaempferol and quercetin showed the same antiepileptic effect level as levetiracetam in the treatment of rat chronic epilepsy model and also showed the regulating pro-inflammatory and anti-inflammatory factors [21]. Formononetin has anti-cancer, hypolipidemic and neuroprotective effects. In a rat model of ischemic stroke, Formononetin has been demonstrated to provide neuroprotection by enhancing neuronal differentiation and synaptic plasticity. β sitosterol and stigmasterol belong to phytosterols, while phytosterol (PSs) cannot be synthesized in mammals and comes entirely from the diet. In addition, PS can reduce cholesterol and lipid-lowering, as well as resist atherosclerosis and immune regulation. PS crosses the blood-brain barrier, regulates lipid-derived pro-inflammatory mediators and plays an anti-neuroinflammation role [22]. It also has the ability to accumulate in the brain [23], which is of great significance for the regulation of neurodegenerative pathways. β -sitosterol (BS) plays an anti-inflammatory role mainly by inhibiting the activation of P38, ERK and NF-κB pathways, and inhibiting the pro-inflammatory mediators in microglia, such as IL-6, COX-2, iNOS and TNF-α [24]. β -sitosterol can fully reach the brain, scour free radicals and inhibit enzymes involved in cholinesterase metabolism to improve movement disorders, short-term memory and cognitive deficits in rats [25]. In addition, it has anti-anxiety [26] and analgesic effects. β -sitosterol improves blood glucose control in type 2 diabetic rat model by activating IR and GLUT4 in adipose tissue [27]. Six Hub genes were screened from Cytoscape, which were AKT1, IL6, VEGFA, CASP3, JUN and MMP9.

Akt1 acts as a cell protector. Regulation by apoptotic mechanisms and activation by survival signaling pathways can provide cellular protection. In a mouse model of ICH, hNSCs overexpressed with Akt1 showed improved survival, differentiation, and animal behavioral function.

IL-6 (interleukin-6) is a pro-inflammatory cytokine and a neuroinflammatory biomarker. In a clinical study of 841 patients, higher IL-6 levels at admission were found to be associated with 90-day adverse functional outcomes, ICH volume, and perihematomal edema [28]. Another group of experiments on animal models of intracranial hypertension showed that increased ICP was significantly associated with increased levels of cytokines, such as IL-6, IL-18, IL-1α and IL-1β [29].

Vascular endothelial growth factor (VEGFA) is an angiogenesis marker that is also critical for nerve repair. In a study of 33 healthy individuals, serum levels of NGF (nerve growth factor) and VEGF were significantly correlated [30]. In addition, VEGFA can affect neuropathic pain. Diabetes can reduce VEGF-A / VEGFR2 signal cascade, leading to spinal cord

endothelial dysfunction and neuropathic pain [31]. Soluble VEGFR1 targeting VEGFA through the AKT/TRPV1 pathway has recently been reported to reduce neuropathic pain [32].

In the study of vincristine on peripheral nerve injury in rats, it was proved that quercetin could provide neuroprotection by inhibiting CASP3 and activating Nrf2 and Akt [33].

Depression after ICH significantly affects prognosis, and Jun is the central gene in the post-ICH depression gene regulatory network [34]. Jun-related products are also involved in inflammation, stress response [35] and the development of central nervous system diseases.

After ICH, the expression level of matrix metalloproteinase-9 (MMP-9) is increased, which is involved in a series of processes after ICH through various mechanisms. It is closely related to prognosis and is a potential therapeutic target [36]. Therefore, we found that the 6 natural core compounds in LTG have multiple effects on blood pressure control, analgesia, inflammation control, lipid-lowering, epilepsy control, depression prevention and treatment after the treatment of ICH by acting on the 6 core gene targets. It is noteworthy that LTG may exert analgesic effect through β -sitosterol. Intracranial hypertension after ICH and craniotomy can lead to headache. Acute pain is associated with surgical complications and adverse outcomes [37]. A study has shown that the implementation of enhanced recovery after surgery (ERAS), which pays more attention to postoperative pain, can reduce postoperative hospital stay and thus improve the recovery rate of patients [38]. A follow-up study of ICH shows that post-stroke epilepsy (PSE) is the only factor that increases mortality among many factors such as age, gender, diabetes, etc. [39]. Kaempferol and quercetin mentioned above have unique anti-epileptic effects and have important clinical value in the treatment of ICH.

Through the GO enrichment results, we observed that the cell components in CC were directly or indirectly involved in the treatment of ICH by LTG. The results of BP and MF have further deepened our understanding of the functions of several core compounds. The results of KEGG pathway showed that the first pathway of LTG that was most likely to affect ICH treatment was AGE-RAGE signaling pathway in diabetic complications. In addition, Lipid and atherosclerosis, Fluid shear stress and atherosclerosis, IL-17 signaling pathway and TNF signaling Pathway play an important role in this pathophysiological process. 59 genes were integrated into the first 10 pathways, among which 6 core genes were interspersed. It also showed that several natural plant drugs in the treatment of ICH had multiple effects through multiple approaches and multiple targets, which was in line with the treatment concept and characteristics of Traditional Chinese medicine prescription.

The results of molecular docking conformation screening confirm that the active ingredients and target proteins of natural phytopharmaceuticals can be predicted by bioinformatics + network pharmacology and computer technology, and the pairs with low binding energy and high affinity can be screened by docking conformation. It indicates that molecular docking technology has some reference value in the development and utilization of natural phytopharmaceuticals.

In the animal experiments, it is particularly interesting to note that an extremely significant difference was shown in the mNSS scores between the NS and LTG groups on postoperative day 1 (p = 0.000), and further increased on postoperative day 5 (p = 0.005) and day 10 (p = 0.001). TTC staining again visually verified that the infarct size was significantly reduced in the LTG-treated experimental group compared to the NS control group. Combined with the progression pattern of ICH, it was further hypothesized that in the early stages, after the development of oedema, the core compound in LTG, through six Hub genes present in multiple KEGG pathways (AKT1, IL6, VEGFA, CASP3, JUN, MMP9), blocked the inflammatory burst and subsequent cascade of destruction in the acute phase of ICH, initiating early multiple programs such as cryoprotection. Its efficacy may have a role in the core problems of acute phase ICH inflammation, perihematomal oedema and intracranial hypertension, while taking

into account multiple aspects such as analgesia, blood pressure regulation, ant epilepsy and neurorepair, especially in the depression of ICH survivors during the recovery period. Thus, it reflects the holistic thinking of Chinese herbal formulations. It may have important clinical value in the acute and mid- and long-term treatment of ICH. With the continuous development of bioinformatics and network pharmacology, the level of individualized treatment of ICH can be improved by adding or subtracting formulations in LTG according to the different comorbidities among individuals in the future development of herbal innovations.

### 4.1 Inadequacies

There are still some shortcomings in this study. (1) Due to the limitation of timeliness and comprehensiveness of gene bank data, the predicted results are deviated from the actual situation. (2) And experiments in vitro cannot reflect the real state of drugs in vivo, so the drug effects are only studied from a certain level. (3) In this study, 6 natural plant drugs in LTG prescription were selected as the research objects. TCMSP is a unique pharmacological analysis platform of Chinese herbal medicine system, and scorpion is not included in this database as an animal drug, so the prediction results may be biased to some degree. (4) The effect of traditional Chinese medicine prescriptions in vivo is the result of the interaction and accumulation of various components, but the study of network pharmacology is relatively simple. (5) Experimental methods should be added later to further verify the predicted results.

## Supporting information

**S1 Data.**
(ZIP)

## Acknowledgments

We would like to thank Professor Jian Guo Xu and Professor Ting-Hua Wang for their guidance. We also thank Prof. Zhong Fu Zuo for his support and encouragement.

## Author Contributions

**Conceptualization:** Jie Sun.

**Data curation:** Jie Sun, Na Li, Min Xu, Li Li, Ji Lin Chen, Yong Chen, Ting Hua Wang.

**Formal analysis:** Na Li, Li Li, Ting Hua Wang.

**Methodology:** Jie Sun, Na Li, Li Li, Ji Lin Chen, Yong Chen, Jian Guo Xu, Ting Hua Wang.

**Resources:** Jie Sun.

**Software:** Jie Sun, Na Li, Min Xu, Ji Lin Chen, Yong Chen.

**Writing – original draft:** Jie Sun, Min Xu.

**Writing – review & editing:** Jian Guo Xu, Ting Hua Wang.

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
