## [Decision Letter · Decision Letter 0]

20 Jun 2022

PONE-D-22-12747Mechanism of gene network in the treatment of intracerebral hemorrhage by natural plant drugs in Lutong granulesPLOS ONE

Dear Dr. Wang,

Thank you for submitting your manuscript to PLOS ONE. After careful consideration, we feel that it has merit but does not fully meet PLOS ONE’s publication criteria as it currently stands. Therefore, we invite you to submit a revised version of the manuscript that addresses the points raised during the review process.

We look forward to receiving your revised manuscript.

Kind regards,

Hongxun Tao

Academic Editor

PLOS ONE

Journal Requirements:

“We confirm that the manuscript has been read and approved by all the undersigned authors.

We confirm that we have given due consideration to the protection of intellectual property associated with this work and that there are no impediments to publication, including the timing of publication, with respect to intellectual property. In so doing we confirm that we have followed the regulations of our institutions concerning intellectual property.

We understand that the Corresponding Author is the sole contact for the Editorial process (including Editorial Manager and direct communications with the office). He/she is responsible for communicating with the other authors about progress, submissions of revisions and final approval of proofs. We confirm that we have provided a current, correct email address which is accessible by the Corresponding Author and which has been configured to accept email from PLOS ONE”

5. Please ensure that you refer to Figure 1 and 5 in your text as, if accepted, production will need this reference to link the reader to the figure.

6. We note you have included a table to which you do not refer in the text of your manuscript. Please ensure that you refer to Table 3 in your text; if accepted, production will need this reference to link the reader to the Table.

Reviewers' comments:

Reviewer's Responses to Questions

**Comments to the Author**

1. Is the manuscript technically sound, and do the data support the conclusions?

Reviewer #1: Yes

Reviewer #2: Yes

Reviewer #3: Yes

2. Has the statistical analysis been performed appropriately and rigorously? 

Reviewer #1: Yes

Reviewer #2: I Don't Know

Reviewer #3: Yes

3. Have the authors made all data underlying the findings in their manuscript fully available?

Reviewer #1: Yes

Reviewer #2: Yes

Reviewer #3: Yes

4. Is the manuscript presented in an intelligible fashion and written in standard English?

Reviewer #1: Yes

Reviewer #2: Yes

Reviewer #3: Yes

5. Review Comments to the Author

Reviewer #1: The research aimed to evaluate the effects of six natural plant drugs in LT on ICH and the relevant mechanisms using bioinformatics and network pharmacology methods. Results showed that LT-6 are beneficial to ICH by participating in the post-ICH process through anti-inflammatory response, nerve repair, analgesia, anti-epilepsy and other aspects. The structure of the paper was well organized, the methods were described in detail and the results were convinced.

Although the paper is in a good quality, there are still some problems should be revised before the acceptance.

1. English language should be revised by native speaker.

2. Please add a space between number and unit. For example, 150 g instead of 150g(Line 157)

3. Line 160: change 100 degrees Celsius to 100 �C

4. Line 165: what is “they” indicates?

5. Line 193: No full name of ICH is needed.

6. What is the meaning of the words in red in Table 2?

7. Line 393: the full name of ICH has been shown at the very beginning of the paper, so no full name is needed here. Please check it through the whole paper.

8. All the figures are not in a high quality, please re-upload them.

Reviewer #2: This is a very complete research paper about the mechanism of gene network in the treatment of intracerebral hemorrhage by natural plant drugs in Lutong granules. However, authors still need to discuss more about the relationship between Results of Animal experiment and gene network

Reviewer #3: In this article, the authors applied network pharmacology methods to elaborate active ingredients and potential mechanisms of LT-6 in ICH. The experimental design is rigorous, but it requires major revisions before it may be suitable for publication. Addressing the below concerns will improve the quality of this research paper.

1. In Line 153-162, the extract yield of Lutong granules is necessary to be shown.

2. Please combine pictures that illustrate the same issue into one figure. For example, Figure 2 and Figure 3 are suggested to be considered as two panels in one figure. Similarly, Fig 4-5, Fig 6-7, and Fig 8-9.

3. Signal pathway and biological progress are at different levels in the disease model. Therefore, It is recommended to rewrite the result with a more logical organization. (part “2.4 Gene ontology (GO) and KEGG signaling pathway analysis” )

4. Please provide the representative pictures of the TTC stain in Fig 10.

6. PLOS authors have the option to publish the peer review history of their article (what does this mean?). If published, this will include your full peer review and any attached files.

Reviewer #1: No

Reviewer #2: No

Reviewer #3: No

---

## [Author Response · Author response to Decision Letter 0]

12 Aug 2022

Response to Reviewers

Journal Requirements:

1.The style of the paper has been modified according to the PLOS ONE style template. 

2.An updated conflict of interest statement has been included with the cover letter.

3.The minimal data has been uploaded as an attachment.

4.ORCID iD. we have provided an ORCID and associated.

5.After checking, all figures and tables are mentioned in the text.

Review Comments to the Author

Reviewer #1:

1.English language should be revised by native speaker.

Answer: Full text has been retouched.

2.Please add a space between number and unit. For example, 150 g instead of 150g(Line 157).

Answer: Already revised(Line 151-152).

3.Line 160: change 100 degrees Celsius to 100 C.

Answer: Already revised(Line 154).

4.Line 165: what is “they” indicates? 

Answer: "They" means SD rats, already revised. (Line 159).

5.Line 193: No full name of ICH is needed.

Answer: Already revised. (Line 195).

6.What is the meaning of the words in red in Table 2?

Answer: List of genes in the top 10 KEGG pathways, where the red genes represent the distribution of the 6 HUB genes in each pathway. (Line238) .

7.Line 393: the full name of ICH has been shown at the very beginning of the paper, so no full name is needed here. Please check it through the whole paper.

Answer: Already revised. 

8.All the figures are not in a high quality, please re-upload them.

Answer: We have re-uploaded.

Reviewer #2

This is a very complete research paper about the mechanism of gene network in the treatment of intracerebral hemorrhage by natural plant drugs in Lutong granules. However, authors still need to discuss more about the relationship between Results of Animal experiment and gene network.

Answer: We have further discussed the relationship between the results of the animal experiments and the gene network. (lines 438-457).

Reviewer #3:

In Line 153-162, the extract yield of Lutong granules is necessary to be shown.

Answer: Already added, The extract yield of Lutong granules is 25%. (Line 156).

2. Please combine pictures that illustrate the same issue into one figure. For example, Figure 2 and Figure 3 are suggested to be considered as two panels in one figure. Similarly, Fig 4-5, Fig 6-7, and Fig 8-9.

Answer: Images have been merged.

3. Signal pathway and biological progress are at different levels in the disease model. Therefore, It is recommended to rewrite the result with a more logical organization. (part “2.4 Gene ontology (GO) and KEGG signaling pathway analysis”).

Answer: Part 2.4 has been rewritten. (Line 206-229).

4. Please provide the representative pictures of the TTC stain in Fig 10.

Answer: Representative images of TTC staining have been provided(Line 290-296).

---

## [Decision Letter · Decision Letter 1]

1 Sep 2022

Mechanism of gene network in the treatment of intracerebral hemorrhage by natural plant drugs in Lutong granules

PONE-D-22-12747R1

Dear Dr. Wang,

We’re pleased to inform you that your manuscript has been judged scientifically suitable for publication and will be formally accepted for publication once it meets all outstanding technical requirements.

Kind regards,

Hongxun Tao

Academic Editor

PLOS ONE

Additional Editor Comments (optional):

Reviewers' comments:

Reviewer's Responses to Questions

**Comments to the Author**

1. If the authors have adequately addressed your comments raised in a previous round of review and you feel that this manuscript is now acceptable for publication, you may indicate that here to bypass the “Comments to the Author” section, enter your conflict of interest statement in the “Confidential to Editor” section, and submit your "Accept" recommendation.

Reviewer #1: All comments have been addressed

Reviewer #2: All comments have been addressed

2. Is the manuscript technically sound, and do the data support the conclusions?

Reviewer #1: Yes

Reviewer #2: Yes

3. Has the statistical analysis been performed appropriately and rigorously? 

Reviewer #1: Yes

Reviewer #2: Yes

4. Have the authors made all data underlying the findings in their manuscript fully available?

Reviewer #1: Yes

Reviewer #2: Yes

5. Is the manuscript presented in an intelligible fashion and written in standard English?

Reviewer #1: Yes

Reviewer #2: Yes

6. Review Comments to the Author

Reviewer #1: (No Response)

Reviewer #2: Authors have adequately addressed my comments raised in a previous round of review and I feel that this manuscript is now acceptable for publication.

7. PLOS authors have the option to publish the peer review history of their article (what does this mean?). If published, this will include your full peer review and any attached files.

Reviewer #1: **Yes: **Ce Shi

Reviewer #2: No

---

## [Editor Report · Acceptance letter]

15 Nov 2022

PONE-D-22-12747R1 

Mechanism of gene network in the treatment of intracerebral hemorrhage by natural plant drugs in Lutong granules 

Dear Dr. Sun:

I'm pleased to inform you that your manuscript has been deemed suitable for publication in PLOS ONE. Congratulations! Your manuscript is now with our production department. 

Kind regards, 

on behalf of

Dr. Hongxun Tao 

Academic Editor

PLOS ONE